# Reciprocal Interactions between Oligodendrocyte Precursor Cells and the Neurovascular Unit in Health and Disease

**DOI:** 10.3390/cells11121954

**Published:** 2022-06-17

**Authors:** Friederike Pfeiffer

**Affiliations:** Department of Neurophysiology, Institute of Physiology, University of Tübingen, 72074 Tübingen, Germany; friederike.pfeiffer@uni-tuebingen.de

**Keywords:** oligodendrocyte precursor cell, vasculature, white matter, gray matter, demyelination, NG2, collagen IV, collagen VI, PDGFRα

## Abstract

Oligodendrocyte precursor cells (OPCs) are mostly known for their capability to differentiate into oligodendrocytes and myelinate axons. However, they have been observed to frequently interact with cells of the neurovascular unit during development, homeostasis, and under pathological conditions. The functional consequences of these interactions are largely unclear, but are increasingly studied. Although OPCs appear to be a rather homogenous cell population in the central nervous system (CNS), they present with an enormous potential to adapt to their microenvironment. In this review, it is summarized what is known about the various roles of OPC-vascular interactions, and the circumstances under which they have been observed.

## 1. Introduction

The blood-brain barrier (BBB) is essential for maintaining homeostasis in the brain by protecting the central nervous system (CNS) from the constantly changing composition of the bloodstream, a prerequisite for proper neuronal function. CNS homeostasis is impaired in many neurological diseases such as multiple sclerosis, stroke, Alzheimer’s disease, brain tumors, and epilepsy [1,2], due to alterations of BBB properties, inflammation, or changes in neuronal excitability. Endothelial cells forming the BBB are in close contact with adjacent pericytes, and astrocytes are ensheathing vessels with their endfeet. It is known that microglial cells and interneurons establish contact with endothelial cells, astrocytes, and pericytes and thus together form the neurovascular unit (NVU) [3]. Neurons are coupled to the NVU through their energy demand: an increase in neuronal activity coincides with blood flow adaptions, which is called neurovascular coupling [4]. Moreover, astrocytes have been shown to participate in the process of neurovascular coupling, too, linking neuronal activity to a local increase in blood flow [5]. Recent studies suggest, that oligodendrocyte precursor cells interact with the structural units of the NVU as well, although the functional impact of this interaction remains to be solved. This review focuses on what is known about interactions between OPCs and that vasculature.

## 2. The Neurovascular Unit

### 2.1. Endothelial Cells

In the brain parenchyma, endothelial cells line the blood vessel lumen and restrict paracellular flux through the tight junctions [6] by forming a true physical barrier. The composition of tight junctions that constitute the blood-brain barrier in the CNS is unique as compared to those that can be found in endothelial cells in the periphery. Brain endothelial cells show great heterogeneity themselves based on their location within the vascular tree [7]. Additionally, the cellular composition at the vessel wall highly varies ranging from small capillaries in the brain parenchyma to large vessels ascending/descending from the meninges. Seven clusters of endothelial were recently described, and differentially distributed based on vascular segment and brain region [8,9].

### 2.2. Mural Cells

Adjacent to the endothelial cells and separated by a basement membrane, mural cells can be found. Mural cells consist of pericytes localized at microvessels (capillaries and venules) and vascular smooth muscle cells (SMCs) that occur in arteries, arterioles, and veins. Pericytes occur much more frequently around CNS vessels as compared to other organs [10] and are a molecularly homogenous population [7] that can appear morphologically different. Pericytes have been shown to be absolutely crucial for regulating BBB properties [11]. Arteriolar smooth muscle cells have been shown to regulate regional blood flow [12]. Pericytes and smooth muscle cells share the expression of the neuron-glial antigen 2 (NG2 or chondroitin sulfate proteoglycan 4, cspg4) with OPCs ([13] and database referred to within).

### 2.3. Perivascular Fibroblast-Like Cells

More recently, perivascular fibroblasts have been characterized that are localized in the perivascular space [7]. Fibroblast-like cells can be found around large arteries and veins (adventitial ECM-producing cells), but their presence around smaller vessels is not clear yet. Sequencing screens revealed a vascular cell population expressing PDGFRα [14]. These cells are probably collagen-expressing fibroblasts [7] localized in perivascular areas and at the meninges [15]. Thus, in addition to OPCs in the parenchyma, PDGFRα-expressing cells can be found in the vascular wall in CNS. These fibroblast-like cells also express NG2 ([13] and database referred to within). Perivascular fibroblasts are supposed to contribute to the formation of basement membranes [16] and fibrotic scars during inflammation [17].

### 2.4. Astrocytes

Astrocytes almost completely cover the vasculature with their endfeet [18,19] and are necessary to induce the BBB phenotype in CNS endothelial cells [20,21]. The loss of astrocyte polarity under inflammatory conditions resulted in BBB dysfunction and edema formation [22].

### 2.5. Basal Lamina

While there is one basal lamina (or basement membrane) localized between the membrane of the endothelial cells and the membrane of astrocyte endfeet at the capillary level, there can be two different basal laminae distinguished at postcapillary venules, the endothelial and the astroglial basal lamina, the latter contributes to the parenchymal basement membrane together with the meningeal epithelial basement membrane (reviewed in [23]. Between endothelial and parenchymal basal lamina, the perivascular space can be found, which is especially prominent during neuroinflammation [24]. The major extracellular matrix molecules of the basal lamina are collagen IV, laminin, nidogen, and heparan sulfate proteoglycans (reviewed in [25]). However, the molecular composition of the endothelial and the parenchymal basal lamina differ: the endothelial basal lamina contains laminin α4 and α5 chains and the parenchymal basal lamina contains laminin α1 and α2 chains [26]. Lamin α2 has been shown to regulate the number of OPCs in the developing white matter [27] and induced oligodendroglial fate choice of adult neural stem cells upon secretion by pericytes [28]. Laminin α2, α4 and α5 chains promoted attachment and migration of OPCs, while α4 and α5 promoted OPC survival [29]. Of note, NG2 has been shown to bind laminin presumably through a domain that is different from its collagen-binding region [30].

### 2.6. Perivascular Macrophages and Juxtavascular Microglia

Perivascular macrophages are considered to be a part of the NVU and reside within the perivascular space in the basal membrane with regionally distinct distribution, sharing the same compartment with perivascular fibroblasts [31]. They most likely contribute to the function and maintenance of the BBB [32], and their location just at the interface between the bloodstream/periphery and CNS parenchyma suggests an involvement in immune surveillance. By contrast, microglial cells reside in the parenchyma and have the possibility to interact with the abluminal surface of the CNS vasculature. Microglia are the resident immune cells of the CNS and play an important role in the structure and function of neural circuits. Under inflammatory conditions, microglia largely interact with blood vessels [33] but recent studies suggest a close association already in the healthy brain [34,35]. A portion of microglia, termed “juxtavascular microglia”, was observed to migrate along blood vessels during early postnatal development [36], especially capillaries in the cortex, but they become stationary once astrocyte endfeet cover the vessels [35].

## 3. Oligodendrocyte Precursor Cells (OPCs)

Oligodendrocyte precursor cells (OPCs) are distributed across the entire parenchyma of the CNS, they have been characterized as their own glial cell population [37]. OPCs are mostly known for their ability to differentiate into mature oligodendrocytes and generate myelin. An interesting feature about OPCs is their capacity for self-renewal which they keep throughout life, but not all OPCs directly generate oligodendrocytes, instead, some do not differentiate for months but remain as OPCs [38,39]. They are often identified by the expression of two membrane-bound proteins: the NG2 antigen (therefore they are also referred to as NG2 cells) and the alpha receptor for platelet-derived growth factor (PDGFRα) [40]. PDGFRα expression in the CNS parenchyma is restricted to cells of the oligodendroglial lineage [41] but can be found along the vasculature (see above). NG2 is expressed on pericytes and smooth muscle cells in addition to OPCs and some RNA expression by endothelial cells [7,13] and thus marks additional cell types localized along the vasculature when labeled (note labeling of vascular structures as well as OPCs in the cortex of NG2DsRed mice in Figure 1A,B).

Our current view is that OPCs exist in different functional states, depending on their location within the CNS and also on the age of the organism (reviewed in [42]). For example, OPCs proliferate and differentiate into mature oligodendrocytes at higher rates in white matter as compared to gray matter in the normal rodent CNS [43,44]. Regardless of these differences, single-cell RNA sequencing has revealed that OPCs appear to be transcriptionally quite homogeneous, while oligodendrocytes appear to be more heterogeneous [14,45].

OPCs have been mostly studied with regard to their ability to detect active neurons, and differentiate into mature oligodendrocytes and myelinate axons [46,47,48]. Nevertheless, given their morphology especially in the gray matter areas of the brain, where OPCs extend numerous processes into the surrounding brain parenchyma (which is why they are often referred to as ‘polydendrocytes’ [49,50]), it will not be surprising to find that they establish contact with other cell types of the CNS. While their interactions with neurons [51] and other glial cells [42] have been reviewed elsewhere, the focus of this review will be on OPC interactions with the vasculature (Figure 1), which can be easily visualized given the vascular expression of some of the most commonly used markers for OPCs. A combination of markers that are expressed by OPCs and/or vascular cells reveals how frequently OPCs are in contact with the dense network of blood vessels in the gray matter (Figure 1C,D).

Evidently, most of our knowledge about OPCs has been obtained by studying rodents, mostly murine, CNS tissue. The limited data available to date on human OPCs suggest that there is a remarkable similarity between human and rodent OPCs with regards to morphological and physiological properties [52]. With newly evolving techniques, e.g., single-cell RNA sequencing, human induced pluripotent stem cells and improved in vivo imaging techniques, more comparisons between species and more data available on human cells in different developmental stages are expected to be available.

### 3.1. Relationship between OPC Processes and Vascular Elements

The functions and underlying mechanisms of OPC-blood vessel contacts are only beginning to be understood. Recent investigations revealed a close relationship and an exchange of growth factors and signaling molecules, which will be summarized here.

Vascular Endothelial Growth Factor (VEGF) is expressed by several cell types in the brain, depending on the developmental stage. Its predominant role is to promote blood vessel formation, but it has effects on other cell types, too [53]. VEGF-A can induce migration of OPCs [19] while VEGF-C was shown to promote the proliferation of OPCs in the optic nerve [54] and after demyelination in the medulla oblongata [55].

While OPCs in co-culture increased the integrity of an endothelial layer by releasing soluble factors other than through the PDGF-BB/PDGFRβ signaling pathway [56], TGF-β1 released by OPCs did support BBB integrity in vitro, and deletion of TGF-β1 specifically in OPCs resulted in severe barrier dysfunction in neonates [57].

Vice versa, trophic factors secreted by endothelial cells such as fibroblast growth factor 2 (FGF2) and brain-derived neurotrophic factor (BDNF) promote OPC survival and proliferation [58].

These findings have strengthened the concept of the ‘‘oligo-vascular niche’’ [58], which extends the original concept of the ‘‘neurovascular niche’’ to include oligodendroglial cells. Moreover, OPCs could play a role in controlling brain homeostasis by inducing blood-brain barrier properties through either direct interactions or secreting soluble factors. In the ‘oligovascular niche’, there is signaling from endothelial cells to OPCs possibly promoting their proliferation, as well as signaling from OPCs that could support vascular remodeling. Several candidates involved in this OPC-endothelial reciprocal crosstalk have been identified by now (reviewed in [59]).

#### 3.1.1. Development

Interactions between migrating glial progenitors from the subventricular zone (SVZ) and growing blood vessels have been recognized long ago in the developing cortex [60], coinciding with morphological differentiation into astrocytes. During development, a cross-talk between neural progenitor cells (NPCs) and endothelial cells was shown to be crucial for inducing the commitment of NPCs towards OPCs during embryonic development [61]. In addition, it could be shown that OPCs use the vasculature for guidance during developmental migration, possibly coordinated with differentiation via the Wnt signaling pathway [62]. Wnt pathway activation thereby led to the upregulation of the chemokine receptor Cxcr4 on OPCs that was then able to bind its ligand Cxcl12 secreted by endothelial cells [62], being an example of a direct endothelial-OPC link. During neonatal angiogenesis, OPCs interacted with tip cells of sprouting vessels in the white matter and OPC numbers correlated with vascular density in the white matter [63]. Vice versa, OPCs do not only form close contacts with blood vessels in the telencephalon, but were moreover proposed to be important for the formation of the vessel network during development, thus influencing those cells forming the vessels [64].

Recently, another gliogenic domain was identified at ventricles in the adult brain: an intraventricular, neural stem cell-derived oligodendrocyte progenitor was shown to be activated in response to injury [65], further highlighting the extent of glial diversity. This discovery also shows that oligodendroglial progenitors interact with the cerebrospinal fluid in certain niches, and possibly integrate signals provided in this special compartment.

#### 3.1.2. Homeostasis

OPCs are more abundant in the developing as compared to the adult CNS. In juvenile mice (P15) OPCs represent 20% of cells in the corpus callosum, and 7% in the cortex. In young adult mice at P60, there is a significant reduction in the percentage of OPCs in the white matter to 4.5%, and a less significant drop to 3% in the cortex (Figure 2A). Thereafter, their number remains constant over the adult lifespan [66].

Concerning their interactions with the vasculature, we recently showed that the contact formation between OPCs and vascular structures was preserved beyond embryonic development and is maintained in young (P15) and adult (P60) mice. Thereby, contacts with the vasculature were not only established by OPCs localized close to the vessels, but also by OPCs localized further into the parenchyma, spanning several µm of distance with their protrusions [67]. In the neocortex, 82% of OPCs had processes that were contacting blood vessels at P15, which increased to 94% at P60. Conversely, 92% of the vascular segments were contacted by OPC processes at P15 and 83% of vessels were contacted by OPCs at P60 [67]. In the white matter, the density of vessels is lower as compared to the gray matter, while the number of OPCs is higher (Figure 2A and Figure 3A–C). We observed that only 50% of OPC in the corpus callosum contacted blood vessels at P15 and 55% at P60 (Figure 2B), probably due to the higher number of OPCs in white matter as opposed to a smaller number of available vessels as compared to the gray matter. Accordingly, almost all blood vessel segments (94%) were contacted by OPCs in the corpus callosum at P15, which decreased to 78% at P60 (Figure 2C). Thus, there are fewer interactions between OPCs and the vasculature in white as compared to gray matter, and therefore fewer opportunities to exchange signals in white matter (for further differences in environmental cues between white and gray matter see below).

What functions could the interactions with the vasculature have beyond the phase of embryonic development? CNS myelination was discussed to be linked to neuronal activity via endothelin expression by the vasculature: reduction in neuronal activity leads to a decrease of vascular endothelin expression [68], a close proximity between OPCs and vessels would facilitate this mechanism controlling the amount of myelin sheaths produced. Additional functions of these interactions remain to be discovered.

#### 3.1.3. Influence of Brain Region

Although the CNS is highly vascularized, the density of vessels highly varies between brain regions (Figure 3). In the cortex of the rat, the capillary density is more than double as compared to Corpus Callosum [69,70], linked to physiological differences such as glucose utilization [71]. This discrepancy results in considerable differences in the portion of OPCs that contacts the vasculature in gray and white matter (see above). Thus, these regional differences in the capillary network could contribute to the regional differences in the functions of OL lineage cells. OPCs are distributed throughout the brain, but their densities can vary depending on the region.

OPC density is higher in white matter as compared to gray matter (see above and [42]), which could be due to the higher need for oligodendrocytes in the white matter. Matching the frequency of contact formation between OPCs and the vasculature, it was shown in white matter that OPCs are frequently localized in close proximity to pericytes, presumably facilitating trophic support of each other [72].

Recently, Nrp1-expression by a subset of microglia in a timely regulated manner in the white matter was demonstrated to promote OPC proliferation during development and during remyelination by transactivating PDGFRα on OPCs [73]. Interestingly, Nrp1 is also highly expressed in blood vessels. Microglia and macrophages, on the other hand, are more abundant in white matter [74,75]. However, it is not known whether these cells present Nrp1 to OPCs as a compensational mechanism when vessels are less abundant in the white matter as compared to the gray matter. Whether OPCs bind to different molecules on different cell types in a regionally and timely specific way still has to be clarified.

Most likely, more regionally specific environmental differences not only related to white/gray matter but also connected to regions e.g., brain versus spinal cord will be revealed, and how these cues affect OPC behavior with regard to the vasculature.

### 3.2. Vascular-Oligodendroglial Interactions in Disease

#### 3.2.1. White Matter Injury

Intriguingly, the first condition to study when thinking about OPCs and the vasculature under pathological conditions would be a stroke model. Since white matter is less vascularized as compared to gray matter and the blood flow is lower, it is more susceptible to stroke (reviewed in [76]). Focal ischemia directly affects oligodendrocytes and myelin [77,78], extending the need for remyelinating therapies to vascular diseases. In a rat model for small vessel disease (SD), changes in BBB architecture were detected before symptoms occurred, and subsequent to these changes OPC numbers increased in white matter due to a block in differentiation and an increase in proliferation [79]. The authors found an increase in heat shock protein 90α (HSP90α) released by endothelial cells, which in turn decreased OPC maturation [79]. Generally, perivascular spaces are larger in the white matter, and enlarged perivascular spaces are observed in SD, and are often associated with white matter lesions and myelin loss [80] although the mechanism behind this link remains to be solved [81]. Interestingly, ischemic white matter damage could be reduced by transplantation of microvascular endothelial cells [82] and the same group later showed a direct effect of these transplanted endothelial cells on increasing OPC survival [83]. After ischemic insult, OPCs secrete matrix metalloproteinase-9 (MMP9) at sites where blood-brain barrier permeability was observed, before demyelination occurred [84]. On the other hand, the number of perivascular OPCs increased after ischemic conditions, secreting angiogenic factors that promoted new vessel formation [85]. Chronic hypoxic injury increased OPC density in white matter and also vascular densities in ferrets [63]. Thereby, OPCs activated the Wnt/β-catenin pathway in endothelial cells in a paracrine way. Transplantation of OPCs in a stroke model had a protective effect on the blood-brain barrier, through activation of the Wnt7/β-catenin pathway [86], although the major source of Wnt growth factors required for BBB maintenance are astrocytes [87]. The canonical Wnt/β-catenin pathway and their ligands Wnt7a/b have been implicated in CNS-specific angiogenesis and vascular differentiation [88,89,90]. Increased production of Wif1 by perivascular OPCs counteracted the effects of Wnt signaling and reduced barrier functions and endothelial cell tight junction integrity during in the context of multiple sclerosis (MS) [91]. In comparison to OPCs, whose numbers markedly decrease with age, pericyte numbers remain fairly stable after birth [72]. OPCs and pericytes are in close proximity to each other in the perivascular region [72] which may facilitate the exchange of soluble factors. After demyelination, pericytes were shown to proliferate and secrete Lama2 (laminin α2), which in turn promoted differentiation of OPCs [92].

PDGF dimers and their receptors are expressed by several cell types at the NVU and some of them have been shown to be involved in BBB regulation [93]. Furthermore, PDGF-C has been shown to be neuroprotective [94]. Recently, blocking PDGFR signaling resulted in an increase in ischemic lesions in the subacute phase of stroke. The BBB protecting effect of PDGF signaling could be related to its binding to PDGFRα expressed by collagen I-expressing perivascular cells [95] possibly inducing TGF-β1 expression in the vascular wall. It remains to be elucidated whether and how the PDGFRα (that can possibly bind to PDGFA, B, AB, or C dimers [93]) expressed by OPCs is involved in BBB regulation.

#### 3.2.2. Inflammation

OPC-vessel interactions have also been studied after myelin damage and during repair. Transplanted OPCs interacted with vessels during their migration towards a demyelinated lesion [96]. Indeed, OPCs seem not only to attach, but to migrate along the vasculature in order to reach areas of demyelinated injury [91]. Under these conditions, OPCs were observed to closely interact with cells of the neurovascular unit [97]. A failure of OPCs to detach from vessels in active lesions resulted in the disruption of the BBB and increased leakiness [91].

Knocking out NG2 in the MS model experimental autoimmune encephalomyelitis (EAE) led to a reduced content of extracellular matrix molecules of the basal lamina around vessels (laminin, collagen IV and VI). But while EAE in wild type mice displayed an increase in vessel-associated OPCs accompanied by vessel leakage, these features did not occur during EAE in NG2 knockout mice [98]. These data point toward a function of the NG2 molecule in linking OPCs, pericytes, and endothelial cells of the BBB.

#### 3.2.3. Multiple Sclerosis

In Multiple Sclerosis (MS), focal demyelinating lesions occur with concomitant neurodegeneration [99]. Lesions are heterogeneous in size, location, and inflammatory cell infiltration [100], but remyelination is possible in both white [101] and gray [102] matter. Interestingly, lesion development differs between gray and white matter, although they occur in parallel during the course of the disease. Generally, white matter lesions show larger infiltrates as compared to gray matter lesions, although the reason for this discrepancy still needs to be investigated [103]. Along these lines, gray matter lesions contain less activated microglial cells and hypertrophic astrocytes as compared to white matter lesions which could be correlated to the content of myelin debris that is apparently higher in white matter [103]. In the white matter, classically T- and B-lymphocytes invade the brain parenchyma, a process that is accompanied by blood-brain barrier damage. There seems to be a slow accumulation of lymphocytes in the meninges and the perivascular Virchow-Robin space, leading to subpial demyelinated areas in the cerebral and cerebellar cortex, associated with diffuse neurodegeneration (reviewed in [104]). Thus, T cells seem to take another route in gray matter as compared to white matter.

Notably, although more OPCs and mature oligodendrocytes are present in normal-appearing white than gray matter, this is reversed when looking at lesions, where there are more OPCs and mature oligodendrocytes in cortical lesions as compared to white matter lesions [105]. Consequently, remyelination is more pronounced in cortical lesions [106]. Whether the higher vascularization and nutrient supply in the gray matter contributes to improving the process of remyelination remains to be determined.

In the healthy brain, immune cell trafficking into the CNS is strictly limited, but important for immune surveillance (reviewed in [107]). During neuroinflammation and lesion formation as it occurs in MS, the permeability of the blood-brain barrier changes. Leakage of nonspecific molecules happens together with extravasation of leukocytes [1], although the sequence of events can vary [108]. Disruptions of the blood-brain barrier thus allow blood proteins to enter the CNS, where they change the microenvironment surrounding the affected vessel. Specifically, the blood coagulation protein Fibrinogen has been demonstrated to enter the CNS via the damaged vasculature in MS lesions, where it inhibited OPC differentiation, thus preventing remyelination [109]. Subsequently, an immune therapy was developed, where an antibody specifically targeting an inflammatory fibrin domain while not interfering with blood coagulation was used to suppress neuroinflammation and neurodegeneration [110]. In chronic experimental autoimmune encephalomyelitis (EAE), OPC accumulated within a 30µm distance around vessels, concomitant with Fibrinogen depositions increasing with time and remodeling of the neurovascular niche at sites of inflammation [111]. Fibrinogen acts via bone morphogenetic protein (BMP) receptor on OPCs to inhibit remyelination [109], thus inhibiting BMP type 1 receptor on OPCs and restoring remyelination in EAE [111]. Another molecule linking barrier leakage to OPC differentiation is cholesterol. Cholesterol is essential for myelin production and also facilitates OPC proliferation and differentiation. Cholesterol is often low in demyelinating lesions, and amongst other mechanisms, it could be lost through the impaired BBB [112]. Thus, changing the milieu around vessels in inflammatory lesions influences OPC differentiation and can influence their potential to contribute to remyelination.

After a nearly complete elimination of OPCs through PDGFRα inactivation, differentiation of Nestin+-progenitors residing in the meninges and perivascular spaces in the brain parenchyma was observed and enables re-population with OPC [15], suggesting that perivascular spaces are important stem cell niches that serve as sources for regeneration, as it has already been described for the subventricular zone [113].

The potential of OPCs to contribute to remyelination in the human adult CNS is particularly interesting with regard to the development of regenerative therapies for MS. Analyzing the integration of ^14^C that has been released during the period of nuclear bomb testing into genomic DNA revealed a much lower rate of generation of new oligodendrocytes in humans as the one observed in mice [114]. Dynamic changes in myelination in the white matter are thus traced back to mature oligodendrocytes that account for myelin remodeling instead of turnover rates oligodendrocytes [114]. These findings raised questions about the actual mechanism of remyelination in humans, specifically whether new myelin is generated by newly made oligodendrocytes or existing ones that survived the destruction of myelin, which is crucial for the development of therapies targeting endogenous regeneration [115]. Along these lines, single-nucleus RNA sequencing in white matter from MS patients revealed heterogeneity amongst oligodendrocytes in the adult human brain with only a few newly formed oligodendrocytes and a reduction of OPC numbers in MS lesions and normal-appearing white matter as compared to control tissue, but also clusters with actively myelinating oligodendrocytes [116]. These findings indicate, that the generation of myelin by already existing oligodendrocytes is indeed possible in human patients affected by myelin loss that could potentially be increased by future therapies.

## 4. The NG2 Chondroitin Sulfate Proteoglycan

What is the function of NG2 on the surface of OPCs and vascular cells? Since the development of antibodies detecting NG2 in the 80s, the molecule was found on immature cells inside but also outside the CNS, specifically in the developing vasculature (reviewed in [117]). Revealing the sequence of NG2 confirmed its nature as a transmembrane protein [118], containing one single transmembrane domain and a short cytoplasmic domain. Out of its three extracellular domains, the D1 domain contains a laminin binding motif [119,120], while the D2 domain is able to bind collagen V and VI [121]. Various extracellular matrix proteins are present in the basement membrane around vessels, produced by different cell types. Vascular endothelial cells express laminin isoforms in the CNS, depending on the developmental stage [23] and missense mutations in collagen IV are associated with defects in vascular stability in several organs including the brain [122]. Thus, there is evidence that NG2 mediates the interaction between the cells it is expressed by and the basal lamina, the membrane-associated lamina around vessels in the brain, through its binding of specific extracellular molecules. This binding results in the anchorage of collagen IV to the cell surface [123], and has also been associated with glioma vascularization and progression (reviewed in [120]). NG2 on pericytes was shown to directly interact with the galectin-3/alpha3beta1 integrin complex on endothelial cells, resulting in an enhanced integrin signaling and an increase in blood vessel development in vivo [124]. During fetal brain angiogenesis, pericytes are in close contact with endothelial cells, and NG2 could be shown to be colocalized with collagen IV [125]. It remains to be determined which molecule at the vessels is targeted by NG2 expressed on OPCs, but it is plausible that NG2 mediates OPC attachment and migration towards the vascular segment. Intriguingly, PDGFRα-positive perivascular fibroblasts appear to express collagen VI, as revealed single-cellell RNA-sequencing [45].

Pericytes frequently approach endothelial cells with their secondary processes at peg-socket contacts [126] and adhesion plaques. The function of these contacts is unclear, but it has been suggested that in the CNS, one pericyte could contact several endothelial cells, thereby integrating their functional behavior [16]. Pericyte-endothelial cross-talk is not well understood, but one of the ligand-receptor pairs involved in this interaction is PDGF-PDGFRβ [10]. Given that they belong to the same receptor family, one could speculate that an OPC-endothelial cross-talk could be implemented by PDGF-PDGFRα signaling pathways. PDGFRα has been recently shown to be protective of BBB function during tissue remodeling in a stroke model [95].

## 5. Conclusions

In summary, evidence is accumulating that OPCs are in close contact with the cells of the NVU (schematic shown in Figure 4), release factors influencing vessel formation and maintenance, and receive trophic support in return. Studying pathologies affecting either the vasculature, the myelin compartment, or OPC themselves, revealed even more reciprocal interactions between those cells. While there is still much to be learned about the regulations and interactions of the NVU and the oligodendroglial cells, global approaches considering all cell types involved will help to shed light on disease mechanisms and possibilities for intervention. Many questions remain to be answered. What is the purpose of OPC-vessel interactions? Is the vasculature mostly used as a guidance cue, a niche for the exchange of trophic factors during development or are OPCs also directly assessing the composition of the bloodstream detecting signaling molecules? In addition, it remains to be determined whether there are regional differences in the extent of close interactions between OPCs and the vasculature and the functional significance of these contacts in the various contexts of the CNS.

## Figures and Tables

**Figure 1 cells-11-01954-f001:**
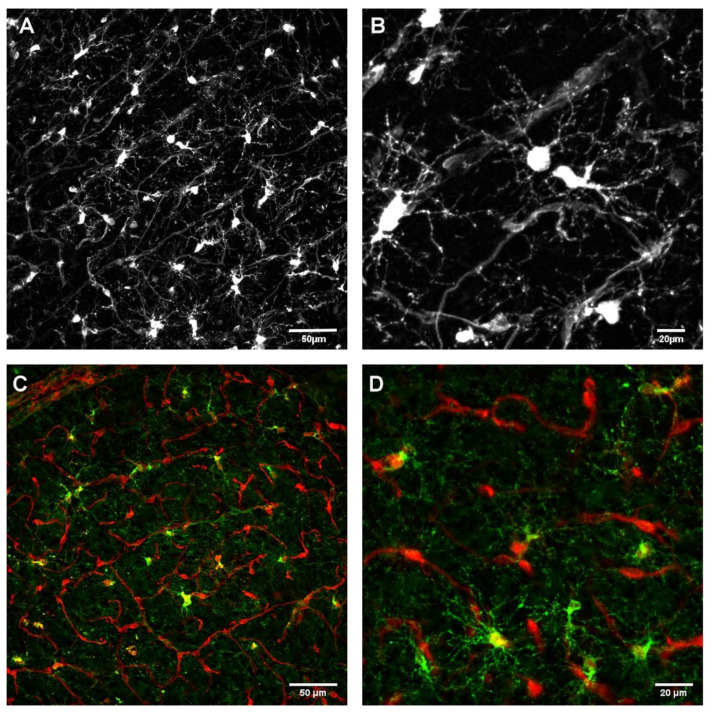
OPCs and the vasculature: numerous contact points are visible in the cortex of adult (P60) NG2DsRed transgenic mice. (**A**,**B**). Expression of a fluorescent marker (DsRed, staining intensified by an antibody detecting red fluorescent protein) under the control of the NG2 promotor reveals its distribution along the vasculature in the dense vessel network and numerous OPCs, distributed throughout the parenchyma in between the vessels. (**C**,**D**). Verification of OPCs by staining with PDGFRα in the cortex of an NG2DsRed mouse (P60). While vascular cells strongly express DsRed (see A and B) and mark the blood vessels in red, OPCs in the parenchyma additionally express PDGFRα (detected with an anti-PDGFRα antibody, shown in green). Therefore, the soma of the OPCs appears in yellow. Note the specific morphology of OPCs as opposed to vascular cells.

**Figure 2 cells-11-01954-f002:**
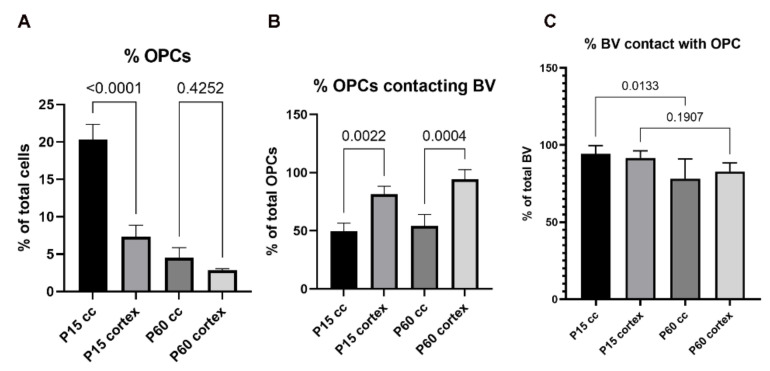
(**A**). Percentage of OPCs per field of view in mouse cortex and corpus callosum compared at P15 and P60. (**B**). Percentage of OPCs that established contact with blood vessels in cortex and corpus callosum at P15 and P60. (**C**). Percentage of blood vessels that were contacted by OPCs in cortex and corpus callosum at P15 and P60.

**Figure 3 cells-11-01954-f003:**
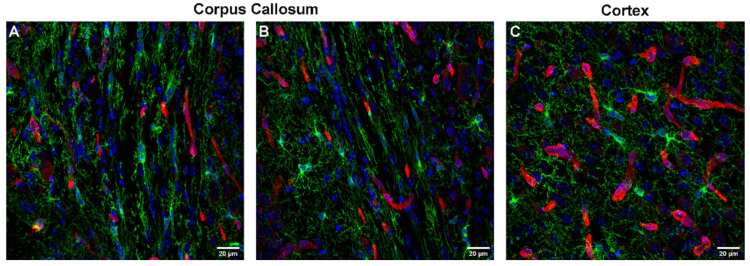
OPCs (green, immunolabeled with a goat antibody detecting mouse PDGFRα) in relationship to blood vessels (red, immunolabeled with a rabbit antibody detecting laminin) in the mouse corpus callosum (**A**,**B**) and cortex (**C**). DAPI staining is shown in blue. Scale bars represent 20 µm.

**Figure 4 cells-11-01954-f004:**
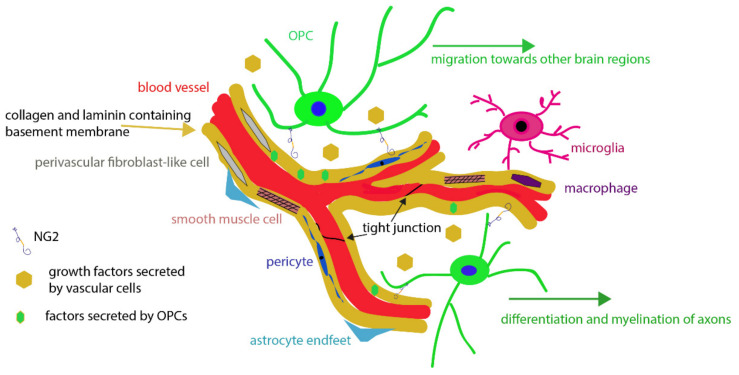
OPCs at the NVU. Schematic showing possible mechanisms of OPC interactions with blood vessels and the different cell types localized at the NVU that OPCs can possibly encounter in this niche.

## Data Availability

Not applicable.

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
