# Peer review of "Reciprocal Interactions between Oligodendrocyte Precursor Cells and the Neurovascular Unit in Health and Disease"

_cells, 2022, doi:10.3390/cells11121954_

Round 1
Reviewer 1 Report
The manuscript entitled "Oligodendrocyte Precursor Cells and the neurovascular unit " is a review article focusing on oligodendrocyte precursor cells and their interaction with the brain vasculature. In addition, the authors have also described the implications of this aspect in vascular diseases of the brain such as stroke and others. The author has also added some results to this manuscript from her own laboratory.
Though this is an interesting topic, this reviewer did not find any novelty in this manuscript, there are a lot of facts and sentences that they are not very well tied together. The title of the manuscript needs to be changed as the title is same as the special issue. The content of the manuscript is not very appealing to the readers and should be made more interesting.
Overall, this manuscript needs major revision and necessary reviewing should be taken into consideration following the comments above. The author needs to clearly identify the topics, the novelty of this specific review and avoid self-citations or addition of recycled results/graphs. If those figures have to be used, they would need to be better incorporated into the flow and rationale of the review.
Author Response
The manuscript entitled "Oligodendrocyte Precursor Cells and the neurovascular unit " is a review article focusing on oligodendrocyte precursor cells and their interaction with the brain vasculature. In addition, the authors have also described the implications of this aspect in vascular diseases of the brain such as stroke and others. The author has also added some results to this manuscript from her own laboratory.
I would like to thank the reviewer for taking the time to assess my manuscript.
Point 1:
Though this is an interesting topic, this reviewer did not find any novelty in this manuscript, there are a lot of facts and sentences that they are not very well tied together.
I worked on improving the flow of the manuscript and added some additional recent findings to it. As this is a review and not an original research article, I am not sure whether I understand what is meant by ‘there is no novelty’ in the manuscript.
Point 2:
The title of the manuscript needs to be changed as the title is same as the special issue.
I changed the title to: Reciprocal interactions between Oligodendrocyte Precursor Cells and the neurovascular unit in health and disease. I hope this makes it more distinguishable from the title of the special issue.
Point 3:
The content of the manuscript is not very appealing to the readers and should be made more interesting.
I tried to make it more interesting.
Point 4:
Overall, this manuscript needs major revision and necessary reviewing should be taken into consideration following the comments above.
I hope the reviewer is content with the changes I made.
Point 5:
The author needs to clearly identify the topics, the novelty of this specific review and avoid self-citations or addition of recycled results/graphs.
I agree I inserted some self-citations about some of our recent work related to the respective topics. While this is frequently done in review articles, I can take them out if the reviewer and the editors decide they do not fit to the topics in the current manuscript. The graphs and images included into the manuscript are all unpublished thus far, although some admittedly similar to what we already published, but I find they nicely illustrate some points on distribution of OPCs with regard to white/gray matter and vessel density, thus, I would like to keep them if possible.
Point 6:
If those figures have to be used, they would need to be better incorporated into the flow and rationale of the review.
I improved the explanations for the figures and increased the connection to the passages where they are mentioned. I think they are better embedded into the text flow now.
Reviewer 2 Report
This is a very well written review authored by a recognized specialist in the field. The topic is new and this review summarizes important and innovative concepts. I think this paper could be improved based on the following comments:
1) The fact that actually 2 basement membranes with distinct laminin compositions are entering in the BBB composition is not mentioned. This is of particular importance since NG2 binds laminins.
2) A cell type of potential importance is not described : perivascular macrophages. These cells, just as pericytes, are localized in the perivascular space i.e in between the basement membranes of the BBB. Importantly, I think the term perivacular microglia is not appropriate although used by many authors. It is a misleading word suggesting that microglial cells can localize in the perivascular cells. I prefer the term "juxtavascular microglia".
3) OPCs are essential for remyelination in rodents but several papers indicate that in humans other mechanisms take place (see for e.g. these articles: https://www.sciencedirect.com/science/article/pii/S0092867414012987
and https://www.sciencedirect.com/science/article/pii/S1084952120301579
See also Jaker et al. Nature 2019, a paper in which human vs mouse molecular differences along the oligodendroglial lineage are deciphered. This paper also demonstrates that the NAWM of MS patients is depleted in specific sub-population of oligodendroglial cells.
These works should be discussed in order to provide a more balanced view on the topic
4) What about NG2 in the human species. It appears that NG2 cells are similar in humans and mice but this should be indicated (https://onlinelibrary.wiley.com/doi/full/10.1002/glia.23725)
5) I think it could be interesting to speculate on potential regional specificities regarding NG2 cells. In particular, it is well established that spinal cord differs from brain with regard to myelination/remyelination processes. What do we know about regional differences (spinal cord vs brain) regarding NG2 cells and their neurovascular niche?
Author Response
This is a very well written review authored by a recognized specialist in the field. The topic is new and this review summarizes important and innovative concepts. I think this paper could be improved based on the following comments:
First of all, I would like to thank the reviewer for the valuable comments, that helped to improve the manuscript.
1) The fact that actually 2 basement membranes with distinct laminin compositions are entering in the BBB composition is not mentioned. This is of particular importance since NG2 binds laminins.
I agree, this fact is important when talking about the basement membrane and NG2 binding. I inserted a passage about the basement membrane (although I used the term basal lamina) under 2.5; line 74-91.
2) A cell type of potential importance is not described : perivascular macrophages. These cells, just as pericytes, are localized in the perivascular space i.e in between the basement membranes of the BBB. Importantly, I think the term perivacular microglia is not appropriate although used by many authors. It is a misleading word suggesting that microglial cells can localize in the perivascular cells. I prefer the term "juxtavascular microglia".
I wrote a passage about microglia and macrophages under 2.6; line 92-106.
3) OPCs are essential for remyelination in rodents but several papers indicate that in humans other mechanisms take place (see for e.g. these articles: https://www.sciencedirect.com/science/article/pii/S0092867414012987
and https://www.sciencedirect.com/science/article/pii/S1084952120301579
See also Jaker et al. Nature 2019, a paper in which human vs mouse molecular differences along the oligodendroglial lineage are deciphered. This paper also demonstrates that the NAWM of MS patients is depleted in specific sub-population of oligodendroglial cells.
These works should be discussed in order to provide a more balanced view on the topic
I added a passage on human remyelination in passage 3.2.3. on MS, line 385-402, and cited those papers. I think this greatly improved the passage.
4) What about NG2 in the human species. It appears that NG2 cells are similar in humans and mice but this should be indicated (https://onlinelibrary.wiley.com/doi/full/10.1002/glia.23725)
This is absolutely true. I inserted a passage concerning the differences/similarities between rodents and humans, line 142-148. Thanks for the comment.
5) I think it could be interesting to speculate on potential regional specificities regarding NG2 cells. In particular, it is well established that spinal cord differs from brain with regard to myelination/remyelination processes. What do we know about regional differences (spinal cord vs brain) regarding NG2 cells and their neurovascular niche?
While this is certainly true, I do not feel very comfortable writing about spinal cord, as I mainly worked in the brain. I added an outlook about different areas of the CNS under 3.1.3. ‘Influence of brain region’, line 276-278. I hope this is sufficient.
Round 2
Reviewer 1 Report
The author answered all the previous queries appropriately.
Reviewer 2 Report
I think this review is a very nice and valuable piece of work. Congrats to the author!